# Centrality of Hygienic Honey Bee Workers in Colony Social Networks

**DOI:** 10.3390/insects16010058

**Published:** 2025-01-10

**Authors:** Adrian Perez, Brian R. Johnson

**Affiliations:** Department of Entomology and Nematology, University of California, Davis, 1 Shields Ave, Davis, CA 95616, USA; brnjohnson@ucdavis.edu

**Keywords:** honey bees, division of labor, social networks, hygienic behavior, social insects, social immunity

## Abstract

Disease outbreaks are a common and important problem in densely populated insect colonies. To combat this issue, workers can perform sanitary tasks, such as dead body removal, which reduce the probability of group-level disease spread but may put the individual at particular risk of contracting or spreading a pathogen. Workers performing these kinds of tasks may, therefore, alter their social behavior or be treated differently by nestmates. We tested this hypothesis by observing the food-sharing interactions of honey bee workers that recently performed tasks involved in removing dead pupae from the nest, an important defense against disease in the honey bee colony. Our results show that workers still take part in the food-sharing network of the colony, even shortly after performing hygienic tasks, and could therefore serve as a source of infection in the colony.

## 1. Introduction

Complex animal societies, such as social insect colonies, can employ both individual and social mechanisms of immunity to prevent, contain, and respond to infection [1,2,3,4]. One important social immunity mechanism in the honey bee colony is hygienic behavior: the uncapping and removal of diseased brood from their cells and the hive [5,6,7,8]. When performed quickly and efficiently, this set of behaviors can underlie colony resistance to a variety of bacterial, fungal, and ectoparasitic pathogens [2,3].

Along with sanitary tasks such as hygienic behavior, colony-level immunity can be achieved through organizational immunity, which refers to temporally dynamic patterns of space use and interaction rates among individuals that can help limit pathogen spread within a colony [1,3,4,9]. Previous work on organizational immunity has provided robust evidence that potentially infectious individuals in insect colonies will modify their social interactions to reduce the probability of infecting healthy nestmates and causing an outbreak [10,11,12,13,14,15]. For example, carpenter ants (*Camponotus aethiops*) infected with a parasitic fungus will eventually cease interactions with other workers and the brood, as well as spend more time outside of the nest [16]. Although previous research strongly suggests that interactions with health-compromised nestmates are often adaptively adjusted, this line of work often lacks detailed information concerning which members of the colony still engage in risky contacts with potentially infectious nestmates and how these interactions fit into the larger web of social connections within densely populated insect nests [17,18].

One approach for more thoroughly quantifying interaction patterns in animal groups is social network analysis, the statistical evaluation of connections among individuals within a social group and how the pathways created by those interactions facilitate or impede the flow of transmittable products such as information, resources, or disease [19,20,21,22]. Using the network toolkit, the social interaction patterns of potentially infectious individuals in the colony can be characterized with greater specificity, both in their direct connections to other nestmates and how pathways of social interactions involving these workers distinguish them as either disproportionally distal or central to the overall colony social network [18]. Furthermore, dynamic network analysis, which incorporates the temporal elements of network connections, can be especially useful for understanding transmission processes like disease spread [18,23,24,25].

Honey bee workers involved in hygienic tasks interact with, or at the very least expose themselves to, dead and diseased brood and are therefore more likely than other within-nest colony members to become infected with various pathogens. For instance, it has been shown that the pupal cannibalism sometimes involved in hygienic behavior can infect workers with deformed wing virus, which can then be spread to other workers via trophallaxis [26]. Likewise, it has long been known that workers in colonies susceptible to American foulbrood remove infected pupae after the bacteria has reached its infectious stage, thus making pathogen intake and transmission likely [6,27]. Workers uncapping cells may also be exposed to pathogens, such as *Nosema* spp. or various viruses, which can be acquired from contaminated wax and then spread through trophallaxis or contact [28,29,30,31,32]. Taken together, it is reasonable to predict that workers recently involved in hygienic tasks may interact less with other nestmates and especially the queen [9,17,33].

In contrast to other sanitary tasks, hygienic tasks necessarily operate on developing pupae and must occur in the brood area where highly valuable newly emerged workers, young nurse bees, and the queen are specifically located [9]. Thus, social segregation, without highly apparent spatial segregation, may be a particularly important solution for reducing contact with hygienic workers. Similarly, if workers recently involved in hygienic tasks are identifiable, it is also possible that hygienic workers mainly share food with each other, either due to being ignored by non-hygienic nestmates in the brood area or due to actively seeking out other hygienic workers. This type of assortative social partner preference is a pattern that can be tested for in social network analyses [22].

Previous studies have operationally defined workers as “hygienic” based on whether they come from a colony specifically bred for high hygienic performance [34,35]. Using colonies with unknown levels of hygienic behavior expression, we identify hygienic bees in this study as a worker that has been directly observed uncapping or removing a dead pupa [36]. Previous work on selectively bred colonies has identified that the propensity to perform hygienic behavior is related to the superior olfactory capabilities of hygienic bees across their lifespan [37,38,39]. Consequently, although most workers have the physical ability to perform uncapping and removing tasks, individuals who actually carry out these behaviors may have correlated differences in traits such as olfactory sensitivity that may also affect their social interactions more broadly [40].

Here, we identify hygienic workers and quantify their direct interactions and the centrality of their position within the colony social network to determine how connected they are to their nestmates via pathways of mouth-to-mouth food exchange (i.e., trophallaxis). We emphasize trophallactic behavior as it is more direct than other social measures such as spatial proximity, and it is a known mechanism of disease transmission between individuals in a colony [41,42,43]. We test four main predictions: (1) hygienic workers should be less central to the colony social network in both their direct connections and their network position; (2) hygienic workers should have fewer direct connections to the queen and young workers; (3) hygienic workers should show preferential interaction with other hygienic workers; and (4) the time-constrained pathways stemming from hygienic workers should be conducive to the containment of a theoretical infection. Given that worker behavior changes considerably with age [44], we primarily test these predictions by comparing hygienic workers to 16-day-old non-hygienic middle-aged bees, which are on average of the same age and caste as workers that perform hygienic behavior [34,35]. Our results provide detailed insight into the social connectivity of workers recently involved in important, but potentially hazardous, social immunity tasks.

## 2. Materials and Methods

### 2.1. Experimental Design

The basic design for this experiment was to establish a colony of individually identifiable honey bees and observe social interactions among nestmates with the novel component of knowing which bees in the colony had recently performed a hygienic task. We assembled 2-frame observation hives in the spring of 2019. Over 5 weeks, we introduced 1500 individually marked workers to a focal hive in weekly cohorts of 300 to establish a colony with a normal age demography. For each cohort, frames of emerging bees were kept in an incubator overnight and then the newly emerged workers were labeled the next day using a paint mark on the abdomen and a plastic tag on the thorax with both a number and color for unique identification. Labeled workers where then introduced to the focal observation at less than 1 day old. At the start of the experiment, we had workers aged 2 days old, 9 days old, 16 days old, 23 days old, and 30 days old for observation.

Focal hives were kept outdoors in a separate field station under a roofed shelter in Davis, California. The morning after introducing the last cohort to a focal hive, we switched out the top frame of the observation hive for a food frame containing mostly open honeycomb, and we switched the bottom frame for a brood frame containing two sections of freeze-killed brood on one side of the frame, a standard method for eliciting hygienic behavior [45]. We manually observed the sections of freeze-killed brood for 4 h and recorded the identities of workers uncapping or removing dead brood. Starting immediately after manual observations, we set up two DSLR cameras (D3200, Nikon, Ayutthaya, Thailand) aimed at either side of the bottom frame and recorded 1 h of footage. We used antireflective glass for improved image clarity and a thin layer of fluon (BioQuip, Compton, CA, USA) to keep bees off the glass and with their tags facing the camera. We performed this experiment two times, with a separate colony and observation hive used for each trial.

We constructed directed and weighted food transfer networks for each colony from the videotape footage. We determined the direction of network interactions by distinguishing which individual donated nectar and which individual received nectar. Edge weights were determined by how long each nectar exchange took place. Interactions under 2 s were not recorded since very short trophallactic interactions rarely involve nectar transfer [46,47]. Every identifiable bee that participated in a food exchange corresponds to a node in the social network. For each individual, we know their age and whether or not they performed hygienic behavior during our experiment, which are coded as attributes assigned to each node in the network. Workers observed performing hygienic behavior during the one hour of video footage were coded as hygienic unless they performed all their social interactions prior to performing hygienic tasks. Using the time at which each interaction occurred, we also constructed time-ordered social networks for dynamic analyses [23,24].

### 2.2. Statistics

All analyses were performed in R version 4.3.3 and R studio [48,49]. Network measures for each colony and for each individual in the networks were produced using functions in the “igraph” package ([50], Table 1). Since social network measures are not suitable for analysis using more traditional statistical methods [9,51,52], we instead used a modified routine specifically designed for network data outlined by Farine [52]. The basic procedure for each analysis is to perform node permutations which randomize labels among individuals to uncouple their phenotype from their network position in permuted networks while also keeping important features of the network, such as the number of each type of individual and the overall structure of the network, consistent. A value of interest is then calculated in each randomized network and in the observed network. The value from the observed network is then compared to the distribution of values from the permuted networks to determine if fewer than 2.5% of randomized values are less than or greater than the observed value (i.e., a two-tailed test for significance in this framework; *p*-values lower than 0.025 are considered significant) [52].

To test hypotheses 1 and 2, which focus on the direct connections between hygienic workers and other nestmates with the queen and young workers considered exclusively in hypothesis 2, we used one thousand node permutations with a restriction in place that selectively randomizes only hygienic worker and non-hygienic middle-aged worker nodes (i.e., nodes from our two focal groups). For the observed network and for each randomized network, we calculated the coefficient estimate for a linear model using the measure being analyzed as the response variable and status as a hygienic bee or a non-hygienic middle-aged bee as the fixed effect. The coefficient estimate in the real network is then the observed value that is calculated and plotted for determining significance as it directly describes differences in the data as opposed to the test statistic or *p*-value [52]. For hypothesis 3, which addresses the social partner preferences of hygienic workers and differently aged workers in the network more generally, we used one thousand node permutations with no restriction in place to randomize the attributes of all nodes in the network and produce permuted assortativity values [53]. In this case, the assortativity values for each network are the values of interest.

We performed the dynamic analyses for testing hypothesis 4 using the “time-ordered” R package [54]. Time-ordered networks explicitly include information on when each interaction occurs such that every pathway in the network represents a biologically plausible transmission route among individuals [25]. Using time-ordered networks, we performed a spread analysis, which determines the proportion of unique individuals in the network that can be reached by a simulated infection initiated from each particular individual in the colony. The simulated spread assumes perfect transmission of the infection. To test the hypothesis that the time-ordered interaction patterns of hygienic workers are conducive to limiting disease spread, we randomly permuted the times at which all interactions in the network took place and then reran the spread analysis for one thousand randomized networks. We then compared the values for the proportion of colony members reached by hygienic workers and non-hygienic middle-aged workers using a linear model as in the previous analyses for the observed network and each randomized network. Essentially, this tests if workers that interact with hygienic workers are in turn ceasing their interactions soon after while workers that interact with non-hygienic middle-aged bees continue to exchange food as normal.

**Table 1 insects-16-00058-t001:** Explanation of network measures.

Measure	Igraph Functions	Description	Source
Degree Centrality	degree: Total exchanges; nectar receptions; nectar donations strength: Total time; time as receiver; time as donor	Number of edges connecting a node to other nodes. Can be directed (giver and receiver are distinguished) and weighted (edges are given an additional value based on the duration of the interaction). Each node’s degree and strength are a sum of all of their discrete connections and the duration of those interactions	[55]
Betweenness Centrality	betweenness	Centrality based on the number of shortest paths between every pair of nodes that pass through the considered node	[56]
Eigenvector Centrality	eigen_centrality	Centrality of a particular node based on its connection to other well-connected nodes. The igraph function uses an adjacency matrix to make this calculation; it is essentially proportional to the sum of centrality measures for all of the nodes connected to an original focal node in each calculation	[22]
Density	edge_density	The number of connections observed in a network divided by the theoretical maximum of connections	[57]
Assortativity	assortativity	Extent to which individuals preferentially interact with individuals of the same attribute (e.g., age, caste). This measure is calculated as the fraction of connections in a network between similarly labeled nodes over the total number of connections in the network	[22]

## 3. Results

### 3.1. General Information on the Colonies and Their Networks

The two colony social networks are visualized in Figure 1 with hygienic workers emphasized as larger, black-colored nodes. Table 2 summarizes the basic structure and properties of the networks. The small values for network density (the proportion of realized connections given the maximum number of possible connections) indicate that both networks are largely unconnected. Table 2 also summarizes information on workers that were observed performing hygienic behavior in the two colonies. For the two tasks of uncapping cells and removing dead brood, the average age of workers involved in hygienic work was between 17.7 and 21.4 days old in the two colonies. In total, 27.4% and 36.4% of workers observed performing hygienic tasks in colony 1 and 2, respectively, were involved in a food exchange during the hour of filming and are, therefore, present in the networks (N = 20 for both).

### 3.2. Are Workers That Perform Hygienic Behavior Less Central to the Colony Social Network?

Table 3 and Table 4 show the network centrality of hygienic workers and non-hygienic middle-aged bees for colonies 1 and 2, respectively. There are no significant differences between the two groups in either their direct interactions or the centrality of their positions in the social network for colony 1. In colony 2, non-hygienic middle-aged bees spent more time giving nectar to nestmates than hygienic workers (*p* = 0.005).

### 3.3. Are Workers That Perform Hygienic Behavior Less Likely to Directly Interact with Younger Workers and the Queen?

Table 5 and Table 6 summarize direct interactions between the queen and workers belonging to the youngest age cohort with hygienic workers or non-hygienic middle-aged bees in colonies 1 and 2, respectively. In colony 1, there are no significant differences in the frequency or duration of interactions between hygienic and non-hygienic workers with the queen and youngest cohort of bees. In colony 2, non-hygienic middle-aged bees had more discrete interaction events, had more discrete interactions in which they donated nectar, and spent more time interacting with these individuals, including both time receiving nectar and time spent donating nectar (See Table 6).

### 3.4. Are Hygienic Workers Positively Assortative?

Figure 2 and Figure 3 show the results of the node permutation tests for each colony for both hygienic and age attributes. Individuals in both colonies are positively assortative based on age and exhibit a degree of assortativity that is significantly greater than expected if age played no role in social partner preference (assortativity = 0.227, *p* < 0.001 for colony 1; assortativity = 0.165, *p* = 0.002 for colony 2). Hygienic workers in colonies 1 and 2 show very slight avoidance and attraction to each other, respectively (assortativity = −0.052 for colony 1; assortativity = 0.033 for colony 2). However, in both colonies, the observed assortativity falls well within the distributions of values from randomized networks (*p* = 0.348 for colony 1; *p* = 0.258 for colony 2).

### 3.5. Do the Full Transmission Pathways of Hygienic Worker Interactions Make Them Less Likely to Serve as a Source of Widespread Infection in the Colony?

We constructed time-ordered social networks for colonies 1 and 2 (Appendix A). These networks show the exact same data as Figure 1 but focus on the timing of network events. Figure 4 shows the results of spread analyses on these networks. The analyses reveal that the maximum reach of any single individual in colonies 1 and 2 is 2% and 2.6% of individuals in the food-sharing network, respectively. Figure 5 shows the difference in spread potential (i.e., unique number of individuals reached at the end of the hour of filming) between non-hygienic middle-aged workers and hygienic workers in our observed network and time-permuted networks. In both colonies, hygienic workers show no significant difference in their spread potential in comparison to non-hygienic middle-aged bees (*p* = 0.262 and *p* = 0.194 for colonies 1 and 2, respectively). In other words, based on the timing of the interactions in pathways initiated by them, hygienic workers and non-hygienic middle-aged bees would spread a novel infection to a similar number of nestmates throughout the colony.

## 4. Discussion

This work provides detailed insight into the social behavior of workers that have recently performed hygienic tasks in the colony and how these workers fit into the greater social network of the colony. Our results from the static and dynamic networks do not generally align with previous studies, which have often shown that infectious or health-compromised individuals will exhibit altered social behavior in comparison to presumably healthy nestmates. We discuss these results within the contexts of colony productivity, organizational immunity, and the general limitations of the experiment.

### 4.1. General Information from the Colonies and Their Networks

The average age of hygienic workers in the two colonies generally matches the results of earlier studies indicating that hygienic workers are usually about 15–20 days old with a considerable amount of variation around that mean [34,35,58,59]. The summary statistics in Table 2 suggest that the two colonies and their networks are largely similar in most respects. The very low density of both networks indicates that they are highly sparse and unconnected. The structure of these networks reflects the fact that these colonies were only observed for one hour, and only a relatively small proportion of workers in a colony will exchange food over small time intervals unless some biologically meaningful process is taking place (e.g., famine relief) [60,61]. Hence, the networks presented here are less dense than networks generated using simpler and more frequent forms of social behavior such as spatial proximity [62,63,64,65] or antennal contact [23], but they may be more relevant for questions concerning potential disease spread since trophallaxis can be a more direct means of transmission for certain pathogens.

Both colony networks contained only a modest percentage of the hygienic workers we observed as well as of the identifiable workers in the colony as a whole. Although it would be worthwhile to perform a baseline comparison of how many hygienic workers do not show up in the network in comparison to how many non-hygienic middle-aged bees do not show up in the network, there are several logistical obstacles involved in obtaining a reliable estimate of the total number of non-hygienic middle-aged bees in the colony (e.g., mortality prior to the experiment and the fact that some non-hygienic middle-aged bees likely spent time on the unfilmed top frame or ramp where they may or may not have been involved in unrecorded nectar exchanges). Despite this limitation, our main focus is on the social interactions that occurred in the brood zone of the colony. We, therefore, concentrate on the network analyses and recognize that we are not sure if hygienic workers were comparatively more or less likely to show up in the networks in the first place.

### 4.2. Are Workers That Perform Hygienic Behavior Less Central to the Colony Social Network?

Results from both colonies largely indicate that workers who recently performed hygienic tasks are no less integrated into the colony network than non-hygienic middle-aged bees either in their direct connections or in the centrality of their network positions (Table 3 and Table 4). There are several possible explanations for this observed lack of difference.

With respect to colony productivity and resilience, it may be that the potential hazards associated with performing hygienic tasks do not outweigh the benefits of remaining integrated in the colony food-sharing network. Highly social insect colonies operate as decentralized systems in which individuals largely use local cues and social information to guide decision-making concerning their task behavior [66,67,68,69,70,71,72,73]. Given that many workers could be involved in hygienic behavior depending on the current state of health in the colony, it would be highly detrimental if most of these workers stopped exchanging food and removed themselves from an important system of resource and information flow in the colony [74,75,76,77]. The ability of workers to shift their behavior as task demand fluctuates is a key component of colony performance [78,79,80,81,82], and workers performing hygienic tasks ought to remain informed on the demand for other tasks. Results from previous studies indicate that persistence in performing hygienic behavior is low for middle-aged bees even in colonies selected for the trait [34,59,83], as well as in situations where demand for hygienic behavior is artificially inflated to a great extent [36]. The notion that workers need to be ready to switch their task behavior in response to new information is of course most relevant for middle-aged workers who are in charge of the majority of tasks inside the nest and who must remain responsive to changes in incoming food from nectar foragers [71,84].

Along with the benefits conferred by remaining in the food-sharing network, understanding the relative risks of hygienic work is equally key for interpreting these results within the context of social and organized immunity. It is important to consider that hygienic behavior encompasses a series of subtasks that are known to be partitioned to various extents based on factors such as colony genetic composition [34,35,83,85]. As in previous studies, we observed variation in worker behavior such that some workers performed both uncapping and removing tasks while others only performed one of these subtasks. This distinction could be meaningful because these two behaviors may pose different levels of risk and immune challenge even if both behaviors likely expose workers to pathogens to some extent. Analyzing uncappers and removers separately would be interesting but would require larger sample sizes and may be more appropriate for future experiments using colonies bred for rapid hygienic behavior where greater numbers of workers performing hygienic tasks can be more quickly observed [34,35,85].

It is also noteworthy that we elicited hygienic behavior in this experiment using dead but not diseased brood as is common in most studies of hygienic behavior. As a result, the probability with which a hygienic worker actually encountered and contracted a pathogen is reduced, and there may be no reason for these workers to show any dramatic differences in how they interact with nestmates as observed in previous studies where workers were directly inoculated with a pathogen [11,12,18] or given some other treatment that compromised their health (but did not make the worker infectious in any way) [13,65,86]. Changes in social behavior from or towards hygienic workers would most likely occur because of an immune response or chemical change in the cuticle that could cue either the hygienic worker or potential social partners to avoid contact [87,88,89,90]. Notably, these changes in behavior and in the cuticle have been shown to occur at timescales similar to those used in this study (e.g., 4 to 6 h after an immune challenge) [91,92]. However, despite previous research showing altered interactions with workers that handle hazardous material but are not necessarily infectious [10], our results suggest that exposure and contact with dead brood are not necessarily sufficient to trigger these kinds of strong responses by hygienic workers or their nestmates. Thus, we cautiously conclude that there are no observable rapid responses to a hazardous stimulus in this caste of bees over the short period of observation we employed.

### 4.3. Are Workers That Perform Hygienic Behavior Less Likely to Directly Interact with Younger Workers and the Queen?

If workers that have recently engaged in hygienic behaviors can host pathogens that can infect other nestmates or develop brood, then it could be particularly imperative that they avoid direct interactions with young workers and the queen [9,26]. We found inconsistent evidence for this prediction over the two trials as only hygienic workers in colony 2 showed significantly fewer interactions with young workers and the queen in comparison to non-hygienic middle-aged bees. Since both groups were equally connected to these particular nestmates in colony 1, we are unable to draw any concrete conclusions as to whether or not this seemingly adaptive strategy occurs in most colonies.

Differences in colony-level results seen here are most attributable to the fact that hygienic workers in colony 2 were particularly unconnected to central nestmates, only receiving nectar from young workers in two short instances and never feeding the queen. A possible explanation may be that genetic and phenotypic differences that underlie the propensity to perform hygienic behavior might also fundamentally affect the social interactions of workers that perform hygienic behaviors regardless of recent task performance, and the extent of these differences may vary at the colony level [37,38,39,58,85]. We compared hygienic workers to workers of the same caste in this study rather than tracking hygienic workers before and after task performance, so it is not clear whether workers who performed hygienic behavior in each colony changed their social interactions after performing sanitary tasks or simply exhibit altered interaction rates with particular nestmates at all times.

Space use is also a fundamental consideration in understanding organized immunity [4]. We also did not record space use and spatial fidelity, which can be fairly compartmentalized even at the scale of a single frame hive [64]. Substantial differences in how often hygienic workers in both colonies spatially coincided with younger workers and the queen could also fundamentally underlie observed disparities in food exchange events between these groups.

### 4.4. Are Hygienic Workers Positively Assortative?

One potential form of social organization in which hygienic workers show no differences in their number of direct connections but are much less central to the colony social network overall is one in which hygienic workers preferentially interact with each other, thereby forming a distinct community that is mostly detached from the rest of the network [20,93]. We tested for this potential social structure using an assortativity metric and found that hygienic workers do not show strong preferences for the type of social partner they interact with (Figure 2a and Figure 3a). Hygienic workers are therefore equally likely to interact with hygienic and non-hygienic workers either because they or their social partners cannot distinguish between the groups or because there is no imperative reason for them to do so. On average, hygienic workers may not have had the necessary contact with a pathogen or other death cue to quickly induce changes that make them recognizable to other workers as potentially hazardous [88,89,90,91].

To verify that our networks still mirrored the basic social structure of a honey bee colony, we performed the same assortativity test based on age. As expected, both colonies show significant positive assortment by age [9,94]. This observed social structure occurs because of the predictable pattern of spatial distribution of different aged workers within the nest and the fact that most social interactions occur within age-based caste groups or between caste groups that work closely together (e.g., middle-aged bees and foragers) [44,64,70,94]. Moreover, it is well established that worker cuticular hydrocarbon profiles vary based on age and caste, and these attributes are detectable by other workers such that consistent interaction preferences are possible even when spatial coincidence between different groups occurs [95,96,97].

### 4.5. Do the Full Transmission Pathways of Hygienic Worker Interactions Make Them Less Likely to Serve as a Source of Widespread Infection in the Colony?

Over the one hour of recorded observation, hygienic workers exhibited a representationally proportional amount of ingoing and outgoing nectar exchanges, as well as time spent giving and receiving nectar as compared to non-hygienic middle-ages bees. Using dynamic network analysis, we were able to probe at the possibility that the time-ordering of hygienic worker interaction pathways may have limited their reach throughout the observed network. The idea here is that hygienic workers may have continued to engage in risky social interactions, but their direct social partners reduced their passage of a potential pathogen in a later interaction by quickly discontinuing their food sharing (i.e., a second-order organized immune response). For the same reasons discussed concerning the results from the static networks above, it is perhaps not surprising that the social partners of hygienic workers did not exhibit any rapid secondary response to contain a theoretical outbreak. To reiterate, there may have not been, on average, a strong response to contact with a hazardous stimulus for these hygienic workers, but a similar experiment with a longer observation period or that used workers directly inoculated with a pathogen or performing some other sanitary tasks might deliver different results.

Again, it is worth noting the limitation of our methods as the disease simulation assumed perfect transmission, and so any nectar exchange with an infected individual led to a simulated pathogen transfer to the naïve individual. Realistically, the probability of disease spread would also correlate with the duration of the interaction, which is considered in the weighted static networks but not the dynamic networks. It should also be repeated that both colony networks are highly unconnected and even the most capable “super spreaders” in both colonies only reached about 2% of individuals in the network over the one hour assessed here. Lastly, we analyze spread based on nectar transfer only and not on contact or other mechanisms of transmission, such as fecal–oral routes, which are beyond the scope of this experiment but could have provided useful insight [42].

The general temporal element introduced here is worth discussing more broadly as most organizational immunity studies that have identified a difference in social or spatial behavior have tested for an effect days or weeks after the treatment is delivered [11,12,14,15,16,89]. Both because hygienic workers are most likely to pick up and spread pathogenic particles directly after performing hygienic tasks and because hygienic workers are already older workers who are soon to transition to their final duties working outside the nest, this study focused on more quickly identifiable differences in behavior. Along with the level of detail that was gathered on interaction partners and interaction pathways involving sanitary workers, the predictions and methods presented here address some important social immunity topics in novel ways that are highly applicable to other immunity-related tasks and pathogen responses. Improvements in methodology, most notably the employment of long-term automated tracking and the use of actually infectious pathogens, would be necessary for fully testing the ideas motivating this work [18,43]. Certainly, there is robust evidence that these kinds of responses prioritizing group-level immunity occur in social insects; however, our work also emphasizes the importance of considering the costs of dramatic responses to hazardous stimuli to group-level performance. Altogether, either due to the importance of middle-aged bees as key within-nest workers with many roles to fill in the colony or due to the particular methods employed here, we find that hygienic workers are no less interactive with a variety of nestmates than non-hygienic middle-aged workers even after very recent encounters with dead brood.

## Figures and Tables

**Figure 1 insects-16-00058-f001:**
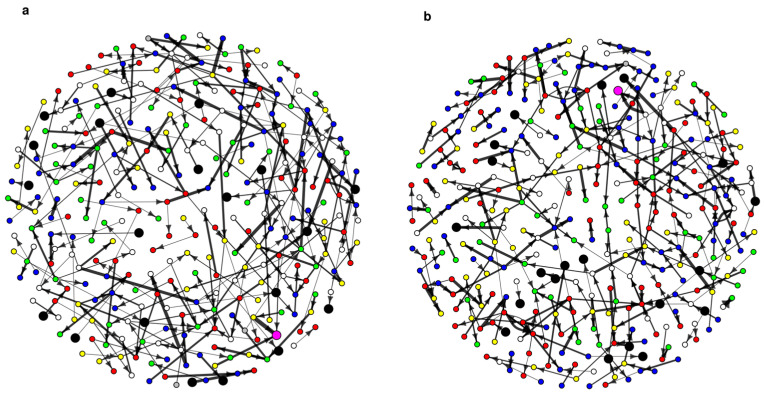
Social networks for colony 1 (**a**) and 2 (**b**). Non-hygienic workers colored by age on the day of the experiment (Blue = 2 days, Red = 9 days, White = 16 days, Green = 23 days, Yellow = 30 days, Gray = unknown age). Hygienic worker nodes are colored black and made larger for emphasis. The queen node is colored magenta and made larger for emphasis. Arrows indicate individual at the base gave nectar to the individual at the tip. Arrow widths are sized based on the log-weight of the duration of the interactions. Note that spatial position in the figure is algorithm-based and not based on spatial behavior of the individual.

**Figure 2 insects-16-00058-f002:**
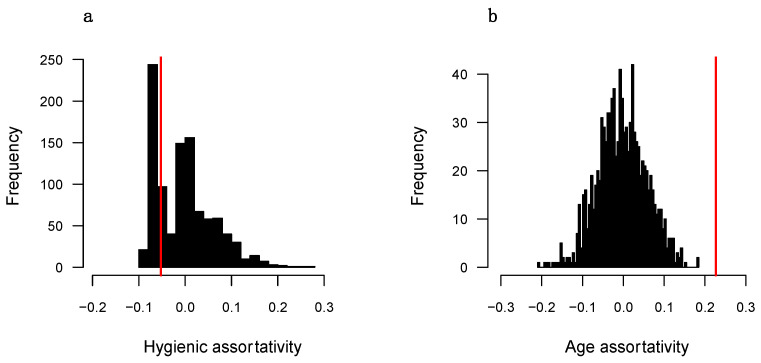
Workers in colony 1 do not preferentially interact based on hygienic status (**a**) but do positively assort based on age (**b**). Shown are the distributions of values from node permutation tests. The red line in each graph represents the observed value from the actual networks. Statistics are reported in the main text.

**Figure 3 insects-16-00058-f003:**
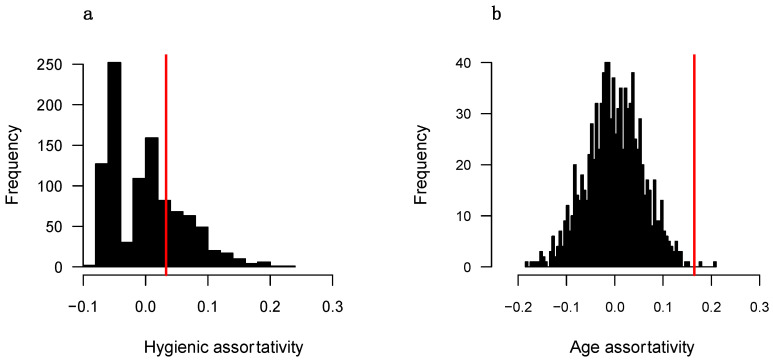
Workers in colony 2 do not preferentially interact based on hygienic status (**a**) but do positively assort based on age (**b**). Shown are the distributions of values from node permutation tests. The red line in each graph represents the observed value from the actual networks. Statistics are reported in the main text.

**Figure 4 insects-16-00058-f004:**
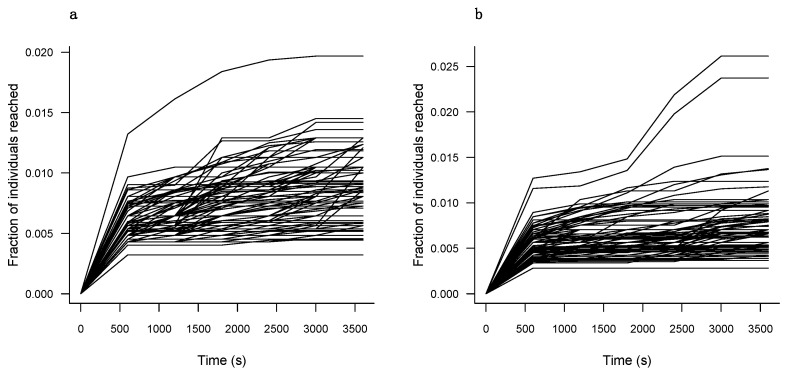
Spread analysis assuming perfect transmission initiating from individuals in colonies 1 (**a**) and 2 (**b**) that serve as an initial source of infection. Each line represents the percentage of individuals receiving a transmittable product from an infection starting from a unique individual in the network.

**Figure 5 insects-16-00058-f005:**
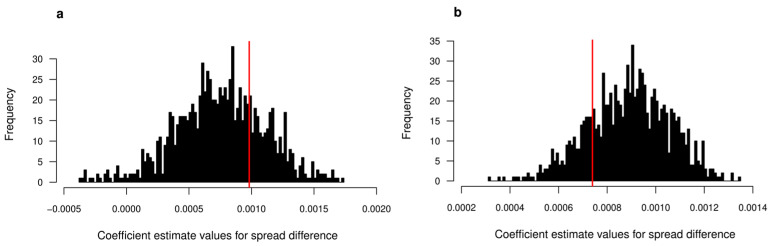
Hygienic workers in colonies 1 (**a**) and 2 (**b**) do not have significantly more or less spread potential than non-hygienic middle-aged workers. Shown are the distributions of values from time permutation tests. The red line in each graph represents the observed value from the actual networks. Statistics are reported in the main text.

**Table 2 insects-16-00058-t002:** Detailed information on colony networks and hygienic workers in each colony.

Measure	Colony 1	Colony 2
Individuals (nodes)	310	354
Trophallactic connections (edges)	258	302
Maximum duration of trophallaxis	170 s	212 s
Network density	0.003	0.002
Total number of hygienic workers	73	55
Uncap	55 (75.0%)	38 (69.1%)
Remove	13 (18.0%)	15 (27.3%)
Both	5 (7.0%)	2 (3.6%)
Mean (±SD) age uncap	17.7 (±9.4) days	21.0 (±8.3) days
Mean (±SD) age remove	21.4 (±7.4) days	19.5 (±8.4) days
Hygienic workers in network	20 (27.4%)	20 (36.4%)
Non-hygienic middle-aged bees in network	59	48

**Table 3 insects-16-00058-t003:** Comparison of interaction patterns of hygienic workers and non-hygienic middle-aged workers in colony 1. The sum of each group’s values is given with the median value in parentheses. Permutation tests give a linear model a coefficient value and a *p*-value based on the randomizations.

		Hygienic (N = 20)	Non-Hygienic (N = 59)	Permutation Test
	Total Nectar Exchanges	26 (1)	99 (1)	0.390; *p* = 0.038
Direct Connections	As Receiver	11 (0.5)	35 (1)	0.302; *p* = 0.321
	As Donor	15 (1)	64 (1)	0.088; *p* = 0.115
	Total Time	323 (6)	1041 (7)	8.288; *p* = 0.309
	Time as Receiver	199 (1)	331 (2)	2.140; *p* = 0.105
	Time as Donor	124 (2)	710 (3)	6.148; *p* = 0.082
Positional Metrics	Betweenness	7 (0)	109 (0)	0.012; *p* = 0.144
	Eigenvector	1 (0)	0 (0)	2.598; *p* = 0.095

**Table 4 insects-16-00058-t004:** Comparison of interaction patterns of hygienic workers and non-hygienic middle-aged workers in colony 2. The sum of each group’s values is given with the median value in parentheses. Permutation tests give a linear model coefficient value and a *p*-value based on the randomizations. Significant values are indicated by an asterisk.

		Hygienic (N = 20)	Non-Hygienic (N = 48)	Permutation Test
	Total Nectar Exchanges	27 (1)	85 (1)	0.378; *p* = 0.06
Direct Connections	As Receiver	16 (1)	37 (1)	0.056; *p* = 0.328
	As Donor	11 (0)	48 (1)	0.321; *p* = 0.034
	Total Time	333 (8)	1508 (16)	10.964; *p* = 0.029
	Time as Receiver	247 (3)	537 (2)	1.126; *p* = 0.308
	Time as Donor	86 (0)	971 (4)	9.748; *p* = 0.005 *
Positional Metrics	Betweenness	13 (0)	45 (0)	0.007; *p* = 0.349
	Eigenvector	0.05 (0)	0.11 (0)	0.703; *p* = 0.328

**Table 5 insects-16-00058-t005:** Comparison between hygienic workers and non-hygienic middle-aged workers in their direct connections to young workers and the queen in colony 1. Permutation tests give a linear model coefficient value and a *p*-value based on the randomizations.

	Hygienic	Non-Hygienic	Permutation Test
Nectar Exchanges	8	16	0.163; *p* = 0.443
As Receiver	3	7	0.136; *p* = 0.489
As Donor	5	9	0.026; *p* = 0.464
Total Time	111	411	6.113; *p* = 0.337
Time as Receiver	80	180	1.709; *p* = 0.157
Time as Donor	31	231	4.405; *p* = 0.130

**Table 6 insects-16-00058-t006:** Comparison between hygienic workers and non-hygienic middle-aged workers in their direct connections to young workers and the queen in colony 2. Permutation tests give a linear model coefficient value and a *p*-value based on the randomizations. Significant values are indicated by an asterisk.

	Hygienic	Non-Hygienic	Permutation Test
Nectar Exchanges	2	26	0.385; *p* = 0.012 *
As Receiver	2	13	0.138; *p* = 0.467
As Donor	0	13	0.247; *p* = 0.006 *
Total Time	4	684	8.592; *p* < 0.001 *
Time as Receiver	4	236	4.187; *p* = 0.024 *
Time as Donor	0	448	4.404; *p* = 0.006 *

## Data Availability

The original data and R code used to analyze the data presented in this study are available in the Open Science Framework repository: https://osf.io/4heuc/?view_only=ed191d5099724f53b68af82e5e26bdd7 (accessed on 1 March 2024).

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
