# Peer review of "Centrality of Hygienic Honey Bee Workers in Colony Social Networks"

_insects, 2025, doi:10.3390/insects16010058_

Round 1
Reviewer 1 Report
Comments and Suggestions for Authors
The work reconstructs the network of relationships between a group of hygienic or non-hygienic middle-aged workers and adult bees and the queen in a honey bee colony.
I have no objections to the results obtained, but I have concerns about whether this experiment is sufficient to verify all hypotheses and form conclusions, such as hygienic bees do not change behaviour, so they are a source of infection. My first question was whether the observation time was adequate to observe the changes.
Therefore, I can agree that the number of contacts of hygienic and non-hygienic bees is similar, while as a source of colony infection, they may not matter, especially when their contact will be with bees older than nurse bees.
This issue is more about the young worker bees, which clean the cells in the first days of life (including after the infected brood) and, in the following days, feed the larvae and potentially spread the pathogen.
Hygienic bees were classified based on the activity of removing frozen brood from the cells, but uncapping and removing cell contents are one of the activities that increases immunity to brood disease.
The social network in the colony is determined by the spatial distribution of different age classes, and the consequence is an organizational structure that provides some immunity to young individuals [17]. Thus, middle-aged bees are not the main source of infection for brood, and their contact with bees of similar age does not necessarily contribute to the spread of infection.
My other doubt concerns the conditions of the experiment. It was assumed that hygienic bees that theoretically become infected during brood removal will modify their behavior to avoid spreading the pathogen. In the experiment, the bees were healthy and did not become infected while removing the frozen brood. They were also not artificially infected. There is no factor (pathogen) that modifies the adaptive behavior of these bees. Some of my doubts are also expressed by the authors in the discussion, in which the authors very reliably verify the results of their studies. This chapter thoroughly discusses all aspects of the research and reliably points out the strengths and weaknesses of the experiment.
The manuscript fulfils the formal requirements and is prepared and written carefully, but some explanations of the study methods and other issues are necessary.
What does it mean the experiment was performed twice? It is not clear to me if two hives colonies were used, or repeated observations on one colony or the hive was settled twice.
Was the frozen brood section under one or two sides of the wax comb?
Does the queen lay eggs? This would help reconstruct the structure of the colony.
Were worker bees labeled and how were individuals in the family identified?
Were workers observed removing dead pupae from the bottom of the hive and were they included in the analyses?
Reviewer 2 Report
Comments and Suggestions for Authors
In this study, the authors employed static social network analysis to quantify the interaction patterns of hygienic workers in the honeybee colony. Their findings indicated no differences in how hygienic workers connected in the social networks. Therefore, they conclude that the honeybee colony lacks mechanisms to protect against hygienic workers spreading infections. However, I have two major concerns that may affect the current conclusions.
1. The experiment was conducted using only two colonies, with a camera recording time of only one hour. This relatively short observation period and limited observation area may not have been sufficient to capture the long-term effects of hygiene behaviors on social networks.
2. For the test of whether health worker bees contribute to the spread of infection, there was no actual pathogen infection or detection in the experiment, which limits the conclusions that can be drawn from the experiment.
Reviewer 3 Report
Comments and Suggestions for Authors
It would have helped the study if baseline data for worker interactions would have been included. Having those data may have shown a change in behaviors from the dead brood challenge.
Reviewer 4 Report
Comments and Suggestions for Authors
Introduction was well-organized and complete. The authors did a good job explaining why the new method would provide insight, which differs from previous research.
The experimental design provides plenty of detail and does a good job mapping out how and why the experimenters made the choices they did.
Line 164: would be helpful to remind the reader of the hypotheses here
Table 1: would be helpful to explain what measures and igraph functions are doing
Fig. 1. Does the duration of the interaction increase the risk of the disease spread? The authors mentioned they did not include short interactions, but past short interactions does the risk increase. If not, the figure would be easier to read if the arrow widths were all the same.
Table 3. Total nectar exchanges p=0.038. Why is this not a significant interaction? Same with Table 4 direction connection as donor and total time.
Discussion is clear and well-organized. The authors did a good job addressing some of the challenges of the experiment and suggestion possible future directions. Although mentioned briefly at the beginning of the discussion, it would be helpful to have more detail about the altered social behavior in health compromised bees and why that was not seen specifically in this study. Likely, because the bees were not infections, but just doing hygienic behaviors.
Although the findings were not significantly different, I do think the paper uses a new technique to explore an important topic in honey bees, disease transmission and has raised some important questions about how we examine hygienic behavior in bees.
Round 2
Reviewer 2 Report
Comments and Suggestions for Authors
My concerns have been addressed.